# Single-Molecule X-ray Scattering Used to Visualize the Conformation Distribution of Biological Molecules via Single-Object Scattering Sampling

**DOI:** 10.3390/ijms242417135

**Published:** 2023-12-05

**Authors:** Seonggon Lee, Hosung Ki, Sang Jin Lee, Hyotcherl Ihee

**Affiliations:** 1Department of Chemistry and KI for the BioCentury, Korea Advanced Institute of Science and Technology (KAIST), Daejeon 34141, Republic of Korea; infinite_l@kaist.ac.kr (S.L.); kihosung@kaist.ac.kr (H.K.); sangj135@ibs.re.kr (S.J.L.); 2Center for Advanced Reaction Dynamics (CARD), Institute for Basic Science (IBS), Daejeon 34141, Republic of Korea

**Keywords:** structure of biomolecules, X-ray scattering, nanoparticle labeling, single molecule, structural ensemble, RNAs, proteins

## Abstract

Biological macromolecules, the fundamental building blocks of life, exhibit dynamic structures in their natural environment. Traditional structure determination techniques often oversimplify these multifarious conformational spectra by capturing only ensemble- and time-averaged molecular structures. Addressing this gap, in this work, we extend the application of the single-object scattering sampling (SOSS) method to diverse biological molecules, including RNAs and proteins. Our approach, referred to as “Bio-SOSS”, leverages ultrashort X-ray pulses to capture instantaneous structures. In Bio-SOSS, we employ two gold nanoparticles (AuNPs) as labels, which provide strong contrast in the X-ray scattering signal, to ensure precise distance determinations between labeled sites. We generated hypothetical Bio-SOSS images for RNAs, proteins, and an RNA–protein complex, each labeled with two AuNPs at specified positions. Subsequently, to validate the accuracy of Bio-SOSS, we extracted distances between these nanoparticle labels from the images and compared them with the actual values used to generate the Bio-SOSS images. Specifically, for a representative RNA (1KXK), the standard deviation in distance discrepancies between molecular dynamics snapshots and Bio-SOSS retrievals was found to be optimally around 0.2 Å, typically within 1 Å under practical experimental conditions at state-of-the-art X-ray free-electron laser facilities. Furthermore, we conducted an in-depth analysis of how various experimental factors, such as AuNP size, X-ray properties, and detector geometry, influence the accuracy of Bio-SOSS. This comprehensive investigation highlights the practicality and potential of Bio-SOSS in accurately capturing the diverse conformation spectrum of biological macromolecules, paving the way for deeper insights into their dynamic natures.

## 1. Introduction

Molecular structures are highly dynamic. The same also holds for biological macromolecules governing the complex vital processes in living organisms. From the helical structures of nucleic acids to the intricate folds of proteins, the three-dimensional architectures of biomolecules are intertwined with the microscopic origins of life. A plethora of both experimental and theoretical studies have consistently demonstrated that biological macromolecules exist as a large ensemble of conformations, rather than adhering to a single static structure [1,2,3]. Small potential barriers between conformations result in continuous thermal fluctuations across various structures, imparting flexibility to biological macromolecules [4,5]. For proteins, this dynamic nature gives rise to the concept ‘protein dynamics’, a topic that has become a notable agenda in the field of structural biology [4,6]. Flexibility in biological macromolecules grants a physiological adaptability essential to the function of biological macromolecules [7,8,9,10,11,12,13]. Representatively, flexible structures facilitate the interactions between different molecules that are essential for processes such as molecular recognition [4,11]. For instance, structural flexibility accounts for the folding of ribonucleic acids (RNAs) [10] or the protein–nucleic acid recognition [7,8]. Furthermore, the capacity of proteins to adapt to the external environment stems from structural fluctuations [13]. Accordingly, a comprehensive understanding of these dynamic features in biological macromolecules is essential for unraveling the complex molecular interplay of life.

Biological macromolecules can adopt multiple structural conformations, each of which may potentially contribute to the physiological functions in vivo. Therefore, to scrutinize the intricate relationship between the physiological function of a molecule and its dynamic structure, it is largely desirable to capture all conformations. On the other hand, traditional structure determination techniques, such as X-ray diffraction (XRD) [12,14,15,16], small-angle X-ray scattering (SAXS) [17,18,19,20], and nuclear magnetic resonance (NMR) [21,22], predominantly capture time-averaged molecular conformations. In these techniques, the structural information is derived from a huge ensemble of biological molecules, each existing in a distinct conformation. If the molecular structure is rigid, meaning that all molecules within the ensemble adopt a single, identical conformation, then these techniques can be employed to retrieve the structure with high spatial resolutions. However, when dealing with biological macromolecules that exhibit structural flexibility, the observables measured from the ensemble average may not accurately represent those of the individual molecules within the ensemble. Thus, directly retrieving the full spectrum of possible structural conformations of the molecule is of paramount importance, yet it presents significant challenges.

To overcome these limitations, a myriad of cutting-edge techniques has been proposed in the fields of XRD [23,24,25], cryo-electron microscopy (cryo-EM) [26,27,28,29,30,31,32], NMR [10,22,33,34], and optical spectroscopy [35,36,37,38,39]. Notably, single-molecule spectroscopy and nanoparticle-assisted cryo-electron microscopy sampling (NACS) [27,32] have emerged as viable strategies for capturing single-molecule structures, each presenting distinct benefits and limitations. Single-molecule spectroscopy is invaluable for capturing molecular behaviors and interactions at the individual molecule level, thus shedding light on the complex and dynamic processes of biomolecules. Techniques such as single-molecule FRET (smFRET) [36,37,38] can elucidate dynamic protein structural ensembles in solution by measuring distances between fluorophores. However, the distance limitation of less than 10 nm for typical fluorophores and the influence of fluorophore orientation can complicate interpretations. NACS has been developed as a complementary approach. NACS utilizes gold nanoparticles (AuNPs) to enhance the visualization of protein conformational distributions. By labeling two distinct sites on a target sample, NACS measures distance distributions, leveraging the principles of cryo-EM. However, it is important to note that cryo-EM involves flash freezing molecules, and the freezing process typically takes several nanoseconds to tens of nanoseconds, during which time molecules may undergo structural rearrangements. This raises questions about the ability of cryo-EM, and by extension NACS, to accurately capture the “instantaneous” conformation of molecules, particularly those that are highly unstable and rarely populated. By contrast, our study employs a novel experimental scheme, called single-object scattering sampling (SOSS) [25], which is based on single-molecule X-ray diffraction (Figure 1a). This technique is tailored to capture and identify individual structural conformations of dynamic biological macromolecules. The SOSS procedure involves irradiating an isolated molecule with a coherent, ultrashort, and high-intensity X-ray pulse. The duration of this X-ray pulse is significantly shorter than the time scale of the typical molecular structural fluctuations, allowing the pulse to capture the conformation of the molecule at a specific instant, as if “freezing” the molecule in time. A repeated measurement of the instantaneous structures allows for sampling all possible structural conformations of the molecule. The relative abundance of each conformation, as determined by repetitive sampling, is then used to reconstruct the population distribution of the structural conformations. In light of these capabilities, SOSS emerges as a promising technique, holding significant potential for investigating the dynamic features of the biological macromolecules.

Nevertheless, two challenges remain in implementing SOSS on biomolecules. First, one should obtain the single molecule scattering snapshot with sufficiently high signal-to-noise ratios in order to retrieve the structural information from the snapshot. However, it is challenging to obtain a scattering image of a large biomolecule that is strong enough to provide three-dimensional structure information with the flux of incident photons from currently available X-ray sources, including the state-of-the-art X-ray free-electron lasers (XFELs) [24,40,41,42]. In addition, for macromolecules such as proteins and nucleic acids, their complex three-dimensional structures, which are composed of more than thousands of atoms, make it practically impossible to extract the structure from a single two-dimensional scattering image regardless of the strength of the scattering signal. One compromise solution to overcome these two challenging obstacles would be the site-directed labeling of biological macromolecules using metal nanoparticles [43,44]. Utilizing nanoparticles composed of heavy elements facilitates the acquisition of clear single-shot single-molecule scattering snapshots [45]. This is particularly useful when two nanoparticles are labeled at distinct sites on a biomolecule, as the distance between these two labeled sites can be routinely determined from the strong interference pattern observed in the scattering snapshot (Figure 1b). Consequently, the site-directed labeling of two nanoparticles enables the extraction of a crucial instantaneous structural parameter—the distance between two labeled sites—from a single scattering snapshot. In this work, we perform proof-of-concept simulations of the SOSS method on nanoparticle-labeled biological macromolecules, a technique denoted as Bio-SOSS. Through these simulations, we demonstrate the feasibility and the potential of Bio-SOSS as a novel approach to probing the dynamic nature of biological molecules.

## 2. Results and Discussion

We present a proof-of-concept of a novel algorithm designed to accurately retrieve the distribution of three-dimensional (3D) interatomic distances within a fluctuating single-stranded ribonucleic acid (ssRNA) chain comprising 2255 atoms (Figure 2a). Each nucleotide in the ssRNA is composed of a ribose sugar, a phosphate group, and one of four bases: adenine (A), cytosine (C), guanine (G), and uracil (U). Owing to its dynamic nature, the structure of the ssRNA incessantly fluctuates, resulting in time-dependent variations in the distances between any two nucleosides of interest. These structural fluctuations were sampled in a total of 100 snapshots generated through molecular dynamics (MD) simulations. For each snapshot, we performed labeling of two selected nucleotides on the RNA with gold nanoparticles (AuNPs). The two sites for labeling were carefully selected to prevent any potential collisions with other atoms in the RNA chain. The details of this selection process are comprehensively described in the Section 3. Consequently, we conducted simulations for a total of 18 pairs of labeling sites. For each snapshot, we generated the single-shot single-molecule X-ray scattering pattern for the RNA labeled with AuNPs. To generate the patterns, we assumed a hypothetical X-ray scattering setup with a two-dimensional (2D) area detector. We then simulated the mock experimental single-object X-ray scattering pattern, S_exp_, as detected by the area detector (Figure 2). To check the dependence of S_exp_ and the efficiency of retrieving the inter-site distances on various experimental parameters, S_exp_ was simulated with varying experimental parameters including the size of the AuNPs, the energy of the X-ray photons, the focal size of the X-ray pulse, and the intensity of the X-ray pulse. By analyzing each scattering pattern, we extracted the 3D distance information (r) between the AuNPs. We assumed that this value corresponds to the distance between the two labeled nucleotides on the RNA, denoted as r*_i,j_*. Repeating this process, we were able to construct a histogram of the distances between the two nucleotides.

An experimental single-object X-ray scattering pattern should contain the interference from all pairs of atoms in the ssRNA as well as AuNPs. In addition, the experimental data contain noise. Considering this, S_exp_ was generated by adding the simulated single-object X-ray scattered image from the AuNP-labeled ssRNA, S_AuNP–RNA_, and the sum of Poisson random noise and uniform random noise at each detector pixel, S_noise_ (Figure 2b). The goal is to extract the distance between two AuNPs from the S_exp_, while the instant structure of ssRNA at the moment of X-ray scattering is not known. The X-ray scattering intensity from a molecule depends on the form factor of constituent atoms, as well as the effects of molecular structures such as interatomic distances. In the case of the former, the atomic form factor increases with an increase in the number of electrons, or the atomic number. Given that the ssRNA chain is composed exclusively of lighter atoms—phosphorus, oxygen, nitrogen, carbon, and hydrogen—the scattering pattern originating solely from the RNA is weaker in comparison to those inclusive of the considerably heavier gold (Z = 79) nanoparticles, as depicted in Figure 2c. Hence, the incorporation of AuNP labels significantly amplifies the scattering signal, thereby facilitating the extraction of precise structural information.

More specifically, the signal S_AuNP–RNA_ can be depicted as follows:
     S_AuNP–RNA_ = S_AuNP_ + S_RNA_ + S_AuNP*RNA_ = S_AuNP, 1_ + S_AuNP, 2_ + S_AuNP, 1*2_ + S_RNA_ + S_AuNP*RNA_
         = |*f*_AuNP,1_|^2^ + |*f*_AuNP,2_|^2^ + 2|*f*_AuNP,1_ · *f*_AuNP,2_| + |*f*_RNA_|^2^ + 2|*f*_AuNP,1_ · *f*_RNA_| + 2|*f*_AuNP,2_ · *f*_RNA_|
= 2|*f*_AuNP,1_|^2^ + 2|*f*_AuNP,2_ · *f*_AuNP,2_| + |*f*_RNA_|^2^ + 2|*f*_AuNP,1_ · *f*_RNA_| + 2|*f*_AuNP,2_ · *f*_RNA_|(1)
where S_AuNP_ represents the scattering signal from AuNPs, S_RNA_ denotes the scattering signal from the RNA, and S_AuNP,1_ and S_AuNP,2_ correspond to the scattering signals from each of the two individual AuNPs. S_AuNP*RNA_ represents the scattering cross-term between the AuNPs and the RNA, and S_AuNP, 1*2_ denotes the scattering cross-term between one AuNP and the other. *f*_AuNP,1_ and *f*_AuNP,2_ correspond to the scattering amplitude of each of the two individual AuNPs, and *f*_RNA_ denotes the scattering amplitude of the RNA. Assuming that the two AuNPs are Identical in terms of their composition and shape, |*f*_AuNP,1_|^2^ = |*f*_AuNP,2_|^2^. When the scattering signal from the AuNPs is significantly stronger than that from the RNA, that is, |*f*_AuNP,1_| >> |*f*_RNA_|, Equation (1) can be approximated as follows:
S_AuNP–RNA_                                           = 2|*f*_AuNP,1_|^2^ + 2|*f*_AuNP,2_ · *f*_AuNP,2_| + |*f*_RNA_|^2^ + 2|*f*_AuNP,1_ · *f*_RNA_| + 2|*f*_AuNP,2_ · *f*_RNA_|                               ≅ 2|*f*_AuNP,1_|^2^ + 2|*f*_AuNP,2_ · *f*_AuNP,2_| + 2|*f*_AuNP,1_ · *f*_RNA_| + 2|*f*_AuNP,2_ · *f*_RNA_|                  ≅ 2|*f*_AuNP,1_|^2^ + 2|*f*_AuNP,2_ · *f*_AuNP,2_| = S_AuNP_(2)

Our strategy involves the assumption that |*f*_AuNP,1_| >> |*f*_RNA_|. Consequently, Equation (2) holds true, allowing for accurately approximating the single-object scattering pattern from an AuNP-labeled ssRNA with that from two AuNPs alone.

To test this idea, we first simulated the single-object X-ray scattering images, S_theo_, using only the scattering signals from the two AuNPs, S_AuNP_, while neglecting the contribution from the fluctuating RNA backbone (Figure 2c). The resulting S_theo_ was then compared to S_exp_, which is the sum of S_AuNP–RNA_ and S_noise_, with the primary contribution coming from S_AuNP–RNA_. The qualitative comparison, as illustrated in Figure 2b,c, confirms that S_AuNP–RNA_ can indeed be accurately approximated by S_AuNP_, given that the principal oscillating features in these patterns are nearly identical in shapes. To quantitatively validate this observation, we conducted further analysis. In this process, the distance r was iteratively optimized to minimize the sum-of-squares of residual at each detector pixel intensity, |S_exp_ − S_AuNP_|^2^, scaled by the error at each pixel. To confirm the validity of our approach, we compared the optimized distance, r_fit_, with the actual distance between the two labeled AuNPs, denoted as r_AuNP_. Our simulation results validate that the optimized interparticle distances, r_fit_, closely mirror the actual distances between the AuNPs, r_AuNP_. Consequently, the reconstructed distribution of r_fit_, denoted as p(r_fit_), obtained through Bio-SOSS, accurately mirrors the true distribution, p(r_AuNP_) (Figure 3). The consistency was observed regardless of the labeling sites.

We quantitatively evaluated the accuracy of the distance retrieval method by calculating the discrepancies between r_fit_ and r_AuNP_ at each snapshot (Figure 4). For this analysis, we gathered 1800 pairs of r_fit_ and r_AuNP_ values corresponding to all 18 labeling-site pairs, as analyzing only 100 pairs specific to a particular nucleotide pair would result in insufficient data. The resulting distributions of the 1800 r_fit_ and r_AuNP_ values, denoted as p(r_fit_) and p(r_AuNP_), are plotted and compared in Figure 4a. A satisfactory agreement between p(r_fit_) and p(r_AuNP_) can be observed. By examining the distribution of discrepancies between r_fit_ and r_AuNP_, denoted as p(r_fit_ − r_AuNP_), we found that the errors in r-values, Δr_AuNP_, follow a Gaussian distribution centered at zero (Figure 4b). This distribution has a standard deviation of approximately 0.23 Å, demonstrating that the structural retrieval error of Bio-SOSS is within this range.

To provide a more comprehensive analysis, we decomposed both r_fit_ and r_AuNP_ into their constituent components, x_fit_, y_fit_, and z_fit_, and x_AuNP_, y_AuNP_, and z_AuNP_, respectively. These components represent the 3D projections of the vector pointing from one AuNP to the other. The comparisons of x_fit_ and x_AuNP_, y_fit_ and y_AuNP_, and z_fit_ and z_AuNP_ are depicted in Figure 4c. Subsequently, we calculated the discrepancies between x_fit_ and x_AuNP_, y_fit_ and y_AuNP_, and z_fit_ and z_AuNP_ to obtain Δx, Δy, and Δz. Decomposing the 3D distance r into its x, y, and z directional components elucidates the key factor that governs the accuracy of distance retrieval. Figure 4d illustrates the distinct widths of the distributions of Δx, Δy, and Δz. It is noteworthy that the width of Δz is comparable to that of Δr_AuNP_, indicating that the accuracy in determining the r value is primarily influenced by the accuracy achieved in quantifying the distance along the z-direction. This can be rationalized by considering the richness of information available in the x and y directions, which lie perpendicular to the propagation direction of the X-ray pulse, in comparison to the z-direction. In terms of the momentum transfer vector q, the maximum magnitudes of q in the x- and y-directions, denoted as q_x_ and q_y_, covered by a detector are substantially larger than the component in the z-direction, denoted as q_z_. Taking the Fourier-transform relationship between r-space and q-space into account, as the maximum achievable magnitude q in q-space increases, the minimum resolvable r in the r-space decreases. Therefore, due to the limitations in q_z_ compared to q_x_ and q_y_, the precision in resolving distances in the z-direction is compromised relative to the x and y directions.

The distributions of Δx, Δy, and Δz were quantitatively evaluated, revealing a marked discrepancy in the breadth of Δz compared to Δx and Δy. For this analysis, we employed Gaussian functions to fit the respective distributions and then extracted the center and width (standard deviation) of the optimum Gaussian fits. Consequently, we found that the center values for Δx and Δy were −4.6 × 10^−3^ and −6.0 × 10^−5^ Å, respectively, which are practically zero. The standard deviations for Δx and Δy were calculated to be 5.3 × 10^−2^ and 5.0 × 10^−2^ Å, respectively, both residing within the atomic scale. Conversely, the center value and standard deviation for Δz, at 1.8 × 10^−2^ and 0.37 Å, respectively, were considerably higher compared to the other two axes. Nevertheless, these values remain smaller relative to the typical pair distances in a macromolecule, which are depicted for the RNA on the horizontal axis of the distribution diagrams for r in Figure 3 and Figure 4a. Such findings underscore the potential of Bio-SOSS in accurately delineating the local structural configurations inherent to biological macromolecules.

To comprehensively evaluate the versatility of Bio-SOSS on a wider spectrum of biomolecules, we conducted a quantitative investigation of its accuracy using ten distinct biomolecules (Appendix A), comprising four RNAs, five proteins, and an RNA–protein complex. Our results demonstrate a notable agreement between the actual distances (r_AuNP_) and those retrieved by Bio-SOSS (r_fit_), as depicted in Appendix A. This level of agreement is comparable to that observed for the MD snapshots of the target RNA (PDB ID: 1KXK). Employing the same experimental parameters as those used to obtain the simulation results shown in Figure 4, the standard deviation of the distance differences, σ(Δr), for these ten biomolecules ranged from 0.097 Å to 0.33 Å (Appendix A). In addition, the results suggest a positive correlation between the σ(Δr) value and the size of the biomolecules, which can be characterized either by the molecular weight or the number of residues (Appendix A). A more detailed discussion on this topic is provided in the “Bio-SOSS on various biomolecules” section of the Appendix A.

To determine the optimal conditions for the retrieval of interatomic distances using the Bio-SOSS method, we quantitatively assessed the impact of the experimental parameters on the statistical distributions of Δr_AuNP_. Six conditions were varied in the simulations: the size of the AuNPs (*R*), the wavelength of the X-ray photons (*λ*), the focal size of the X-ray pulse (*f*), the intensity of the X-ray pulse (*I*), the sample-to-detector distance (*l*), and the number of pixels in the detector (*p*). It should be emphasized that *f* and *I* of the X-ray pulse are interrelated, as they collectively define a key independent variable—the number of X-ray photons per unit area. Despite this interconnection, it is essential to recognize the capacity to independently adjust these parameters during the experimental procedure. In light of this, we investigated the effects of varying the two parameters to comprehensively understand their impact on our results.

Scattering images were generated under a variety of experimental conditions, and from these images, the 3D distances, r_fit_, were subsequently extracted. To isolate the influence of each parameter, we systematically varied one parameter at a time while keeping the other parameters constant. The fitted distance, r_fit_, for each image was compared to the actual distance, r_AuNP_. Subsequently, the distributions of discrepancies between r_fit_ and r _AuNP_, Δr_AuNP_ = r_fit_ − r_AuNP_, were generated and fitted with a Gaussian function (Appendix A). The standard deviations of the Gaussian functions obtained from the fitting results, plotted according to conditions, are presented in Figure 5 and Figure 6. We decomposed both r_fit_ and r_AuNP_ into their respective x, y, and z directional components and performed a comparative analysis. The deviations in these components, denoted as Δx, Δy, and Δz, were quantitatively assessed by fitting their respective distributions to Gaussian functions and subsequently determining the center and standard deviation of the best-fit Gaussian functions. The standard deviation values of Δx, Δy, and Δz, along with the standard deviation values of Δr_AuNP_, are collectively displayed in Figure 5 and Figure 6.

One of the most crucial parameters governing the accuracy of Bio-SOSS is the size of the AuNPs used for labeling, denoted as *R*. To quantify the effect of *R*, we traced the dependence of the standard deviation of the difference distribution, σ(Δr), at five distinct *R*s, namely 6, 7.5, 9, 10.5, and 12 Å. We found a monotonic decrease in σ(Δr), or an enhancement in the Bio-SOSS accuracy, as *R* increased (Figure 5a). These results can be attributed to the positive dependence of the AuNP form factor, *f*_AuNP_, on *R* (Figure 5b), which qualitatively accords with the tendency in Figure 5a. As *R* increased, the relative contribution of X-ray scatterings from AuNPs became more significant compared to those from the fluctuating RNA backbone. On the other hand, it is noteworthy that when *R* became large, a disparity between the two AuNPs and the actual distance between the two labeled sites on the biomolecule also arose. This potentially poses negative impacts on the accuracy of Bio-SOSS. Furthermore, for the practical implementation of Bio-SOSS, it is essential to consider both chemical and physical interactions between the AuNPs and the target biomolecule to be labeled. Larger nanoparticles, particularly those with substantial surface charges, are prone to stronger interactions with the target biomolecule, potentially leading to distortions in its native conformation [46]. Since the primary objective of the Bio-SOSS method is to retrieve the native conformation of the biomolecule, rather than the altered conformation due to AuNPs, it is generally less preferable to use nanoparticles of large sizes during the experimental labeling process. In this regard, the choice of metal for nanoparticle labeling can also be of critical importance. For example, AuNPs offer several advantages: (1) they can be synthesized in small sizes, (2) their surface charges can be tailored to desired levels, and (3) they exhibit weak oxidative properties. Collectively, these characteristics make AuNPs superior candidates for Bio-SOSS applications. It is important to note that our simulated labeling process and the results presented in Figure 5 do not take these chemical and physical interactions into account. Instead, our primary aim is to demonstrate the Bio-SOSS method’s capability to retrieve distances in biomolecules, regardless of whether the labeled molecule undergoes denaturation. In practice, the successful application of Bio-SOSS relies on selecting an optimal AuNP size that strikes a balance between the effects of scattering interactions, as discussed in this paragraph (where a larger *R* is advantageous due to the increased scattering contribution from the AuNPs), and the effects of chemical and physical interactions (where a smaller *R* is preferable to minimize distortions from the native conformation).

The X-ray wavelength, or energy, used for the scattering experiment directly determines the maximum range of q (q_max_) that encrypts the structural information of a molecule. Since the reciprocal (q) and radial (r) spaces are interrelated by Fourier transformation, the scattering data with larger q improve the accuracy of structural retrieval at the entire radial space. Therefore, it is commonly assumed that a higher X-ray energy is more beneficial in X-ray scattering experiments. However, our observation in Figure 6a counters this intuition, as the standard deviation of Δr, σ(Δr), increases as the X-ray energy increases. In other words, using an X-ray at higher energy is proven to be disadvantageous for the accuracy of the Bio-SOSS. To assess these results, we calculated the radial resolution limit r_lim_ at each X-ray condition by implementing the well-known relationship r_lim_ = 2π/q_max_, and obtained: 6.7, 4.4, 3.3, 2.2, 1.7, 1.3, and 1.1 Å (at 3, 4.5, 6, 9, 12, 15, and 18 keV, respectively). All of these values are notably larger than the extracted standard deviations of Δr, implying that the additional information obtained from the lengthened q range is not crucial to the functioning of the Bio-SOSS structure. On the other hand, when we computed the dq value corresponding to a single pixel at the center of the detector, we obtained the values 0.009, 0.013, 0018, 0.027, 0.036, 0.044, and 0.053 Å^−1^ at 3, 4.5, 6, 9, 12, 15, and 18 keV, respectively, which are increasing according to the X-ray energy. Therefore, we conclude that the primary factor determining the statistical accuracy of Bio-SOSS is not the maximum range of q-space but rather the q-space resolution corresponding to a single pixel in the detector. It is also expected from the form factors of an AuNP (Figure 5b) where *f*_AuNP_(q) at smaller qs are notably greater by several orders in a logarithmic scale than those in larger qs. To systematically look into the impacts of the two countering effects, we investigated the statistical dependence of the Bio-SOSS accuracy on the size of a 2D area detector. Considering that the pixel size is kept at a constant value of 234 μm, a larger detector increases the maximum available q range while maintaining the single-pixel dq as constant. Figure 6b confirms that the standard deviation of Δr remained almost constant as the number of detector pixels increases. This observation coincides with our hypothesis that the single-pixel dq, rather than the maximum q value, primarily decides the accuracy of Bio-SOSS.

We also observed how σ(Δr) depends on *I* (Figure 6c). A straightforward tendency was observed up to 10^16^ photons, where brighter X-ray reduced σ(Δr), improving the accuracy of the Bio-SOSS. On the other hand, when we used an ultra-intense X-ray with a 10^17^ photon count, the accuracy of the structure retrieval between the two AuNPs significantly worsened. This is because the scattering image from these ultra-intense X-ray sources is sensitive not only to the distance between two AuNPs but also to the delicate structures of RNA. Thus, at these high fluences, we expect that the structural resolution of the RNA backbone becomes high enough to resolve the light atoms at that cost of reducing the distance retrieval accuracy between the two AuNPs. Nevertheless, considering that a typical X-ray photon flux for the X-ray scattering experiments at X-ray linear free electron lasers (XFELs) lies between 10^12^ and 10^13^, we can assume that the accuracy of Bio-SOSS will increase at stronger fluences within the experimental conditions available in near future.

The *f* at the point of interaction also plays a critical role in the spatial resolution of the reconstructed molecular structure. If the X-ray pulses are spatially broad, the intensity per unit area diminishes and the overall scattering intensity decreases. On the other hand, if the X-ray pulses are too tightly focused, the scattering contribution from the RNA backbone and other minor distances becomes non-negligible. In this case, the X-ray scattering images contain mixed structural information not only from the two AuNPs but also from the RNA backbone, as in the case where we use the X-ray pulses with 10^17^ photon flux. The standard deviation of Δr as a function of the X-ray focal size (Figure 6d,e) portrays the combined effect of these two contradicting dependencies, leading to a U-shaped optimal range of focal sizes from 3.2 to 100 nm. Finally, the trend of σ(Δr) with respect to *l* (Figure 6f) shows a combined effect of (1) maximum q, (2) single-pixel dq, and (3) X-ray intensity per unit area. As *l* lengthens, all of these three dependent variables decrease. Here, the increase in single-pixel dq resolution accounts for the initial enhancement in σ(Δr) from 20 to 40 mm while the reduction in unit X-ray intensity predominantly contributes to the decay of the Bio-SOSS accuracy from 40 mm to 80 mm.

## 3. Materials and Methods

### 3.1. Characteristics of Biomolecules Used for Bio-SOSS Simulations

We used four RNAs, one RNA–protein complex, and five proteins for our Bio-SOSS simulations. The selected RNAs were: (1) the rRNA of the HIV-1 virus (PDB ID: 1EBQ, 9.36 kDa) [47], (2) the yeast ai5g group II intron (PDB ID: 1KXK, 22.67 kDa) [48], (3) the tRNA for the aspartylation mechanism (PDB ID: 6UGG, 49.63 kDa) [49], and (4) the adenosylcobalamin riboswitch (PDB ID: 4GXY, 61.46 kDa) [50], serving as representative examples. Additionally, we included a ribosomal RNA–protein complex (PDB ID: 1MMS, 70.43 kDa) [51] in our simulations. The proteins selected for simulation included: (1) the first WW domain of human Smurf1 (PDB ID: 2LB0, 5.32 kDa) [52], (2) photoactive yellow protein (PDB ID: 2PHY, 14.05 kDa) [53], (3) myoglobin (PDB ID: 1MBN, 17.87 kDa) [54], (4) a single unit of green fluorescent protein (PDB ID: 4KW4, 27.23 kDa) [55], and (5) hemoglobin (PDB ID: 2HHB, 64.74 kDa) [56]. This diverse group of biomolecules highlights the versatility of the Bio-SOSS method. The structures of these biomolecules are shown in Appendix A, along with details such as molecular weight, total number of atoms, and the number of bases or residues. Initially, we employed the Bio-SOSS simulations on single static structures of these biomolecules in various orientations to verify the method’s capability in retrieving distances between the AuNPs. Additionally, we conducted molecular dynamics (MD) simulations specifically on the yeast ai5g group II intron RNA (1KXK) to capture a range of dynamic structural conformations. The application of Bio-SOSS simulations to these diverse conformational snapshots effectively demonstrated the capability of Bio-SOSS in revealing the complex and dynamic structures inherent in these biomolecules.

### 3.2. Labeling of Gold Nanoparticles and Generation of Orientational Diversity in RNAs, Proteins, and an RNA–Protein Complex

To obtain diverse molecular orientations for the ten biomolecules (Appendix A), we utilized the Euler rotation method. This method allowed us to introduce orientational variations into the three-dimensional coordinates of the biomolecules without altering their relative connectivity or size. Starting with well-established PDB structures [47,48,49,50,51,52,53,54,55,56], we generated 30 different sets of Euler angle pairs (α, β, γ) at random. By applying each of the 30 distinct Euler rotations to the original structure obtained from the PDB, we generated 30 unique cartesian coordinates for each biomolecule. These sets maintained the molecular conformation of the biomolecules but exhibited different orientations concerning the laboratory frame (the frame defined by the direction of the propagation of incident X-ray pulses and detector alignment, as described in Figure 2a). Within each of these structures, we introduced two AuNPs with a specific radius denoted as *R*. Specifically, we selected two oxygen atoms of the phosphate group (“O1P” and “O2P”, also denoted as “OP1” and “OP2” in some PDB files) for RNAs and the oxygen atoms involved in the peptide bond (–CO–NH–) for proteins as the labeling positions. From the available positions of AuNP pairs, we randomly selected between two and six pairs of AuNP positions, ensuring that their inter-AuNP distances, denoted as r_AuNP_, covered a broad range of distances within the limits allowed by the respective biomolecule in our simulations. To achieve this, we applied a constraint requiring a minimum difference of 10–15 Å in the r_AuNP_s for the selected pairs. We imposed an additional constraint that the distance r_AuNP_ should surpass 2.4 times the radius *R* to avoid the proximity of the two AuNPs. Further details can be found in Appendix A. Appendix A provide a summary of the sizes of the ten biomolecules, alongside the minimum and maximum r_AuNP_, the number of tested pairs, and the corresponding σ(Δr) values obtained from the Bio-SOSS simulations for each biomolecule.

### 3.3. An Application of Single-Object Scattering Simulation (SOSS) to Biological Molcules

The proof-of-concept simulation of Bio-SOSS comprised four key steps: (1) preparation of atomic coordinates; (2) computation of X-ray scattering snapshots; (3) retrieval of the distance between two gold nanoparticles attached to the two positions of interest; (4) statistical sampling of the distance distribution and evaluation of their accuracies.

As the primary step, we prepared the atomic coordinates of the AuNP-labeled biological macromolecule fluctuating freely within space. Structural snapshots of this fluctuating molecule were theoretically obtained by molecular dynamics (MD) simulations. The AuNPs were attached to randomly selected pairs of two atoms in a molecule, with the only constraint ensuring no collision between the other existing atoms over the entire set of snapshots. A detailed account of the MD simulation procedure and the attachment algorithm of AuNPs to the molecular snapshots is provided in the Appendix A.

Subsequently, hypothetical X-ray scattering snapshots, S_exp_, for the AuNP–biomolecule complexes were calculated by using the instantaneous structure of the complex prepared in the first step. In the calculation, the scattering was assumed to be measured on a two-dimensional area detector using the following equation.
S(*u*, Ω) = 0.5 (1 + cos^2^ 2*θ*) · Ω · r_e_ · ∫ I(*t*) · Σ*_j_ f_j_*(*t*) · |exp^iΔk(***u***)·x*j*(*t*)^|^2^ d*t*(3)

In this equation, *u* is the position vector of a detector pixel, Ω is the solid angle of the detector pixel, *θ* is a half of the scattering angle corresponding to the pixel, *r*_e_ is the classical radius of an electron, I(*t*) is the instant photon flux at time *t*, *f_j_*(*t*) is the atomic form factor of the *j*th atom at time *t*, Δk(*u*) is the change in the scattering vector, and x*_j_*(*t*) is the position vector of the *j*th atom at time *t*. This equation calculates the number of elastically scattered photons. Incoherent scattering is ignored in this calculation. To calculate the scattering intensity detected at each pixel, we considered the intensity only at the center of the pixel and assumed that the intensity is equal at all different positions within one pixel. The form factor of a nanoparticle was approximated as follows [57].
*f*_AuNP_(q, *R*) = 3 *f*_Au_(q) · [sin(q*R*) − q*R* · cos(q*R*) ] · q^−3^(4)

Here, q is the magnitude of the momentum transfer vector between the incident and elastically scattered X-ray photon, *R* is the radius of the nanoparticle, *f*_AuNP_ is the X-ray scattering form factor of the nanoparticle of radius *R*, and *f*_Au_ is the X-ray scattering form factor of a gold (Au) atom. The detailed parameters and considerations for the calculation of X-ray scattering snapshots are listed in the Appendix A.

The third stage involves the retrieval of the distance between the two positions of interest where the two AuNPs are labeled. The theoretical image, S_theo_, was simulated assuming that the scattering solely comes from the two AuNPs. The three-dimensional distances between the two AuNPs, namely x_fit_, y_fit_, and z_fit_, were iteratively optimized to minimize the least-squares difference between the mock experimental and simulated images, |S_exp_ − S_theo_|^2^, scaled by the pixel errors. An in-depth description of the fitting procedure is provided in the Appendix A.

The least-squares minimization was repeatedly conducted to obtain a set of x_fit_, y_fit_, and z_fit_ for each hypothetically generated experimental X-ray scattering snapshot. These extracted values were compared to the distance between the two sites of interest, namely x_AuNP_, y_AuNP_, and z_AuNP_, obtained directly from the MD snapshots that were used to generate the theoretical X-ray scattering data, S_theo_. Finally, we employed a Gaussian distribution function to describe the distribution of the difference between the distances from the SOSS and those from the MD snapshots, Δx, Δy, Δz.

### 3.4. Molecular Dynamics Simulation

To generate the various model structures for RNA, we performed the MD simulations using a crystal structure from the yeast ai5g group II intron (PDB ID: 1KXK) [48] as the initial structure. The simulations were performed using the GROMACS 2018 package with the Amber03 force field, in combination with the SPCE water model [58]. The system was equilibrated under the NVT condition for 100 ps with a velocity-rescale thermostat (τ_T_ = 2 fs and T = 300 K) and was subsequently equilibrated under the NPT condition for 500 ps with a velocity-rescale thermostat (τ_T_ = 2 fs and T = 300 K) and a Parrinello–Rahman barostat (τ_T_ = 2 fs and T = 300 K). Following the equilibration, we conducted 5 ns production simulations on the equilibrated structure under both 300 K and 400 K conditions. After the simulations, model structures were sampled at 50 ps intervals from each MD trajectory.

## 4. Conclusions

In this work, we introduced a modified version of SOSS, denoted Bio-SOSS, to capture the array of structural conformations of biological macromolecules. Our systematic investigation has evaluated the plausibility of the Bio-SOSS method, shedding light on the influence of critical parameters such as AuNP size, X-ray energy, and other pivotal factors on its accuracy and reliability. Through a rigorous and systematic exploration, we have pinpointed the optimal conditions essential for obtaining precise measurements, thereby laying the groundwork for future realization of the Bio-SOSS methodology. The insights gleaned from our study are crucial for the continuous refinement of the Bio-SOSS method, significantly enhancing its capability to capture the structure of various conformations, including the rarely populated states, that are intricately linked with complex biological processes. The Bio-SOSS method, with its exceptional structural sensitivity and its ability of capturing rarely populated states, holds immense potential to revolutionize our methodologies for deciphering complex biological processes. As we continue to delve into the capabilities of Bio-SOSS, we expect that this method will unveil intricate details of the dynamic structures of biological macromolecules. This, in turn, is expected to substantially enhance our overall comprehension of vital biological phenomena, thereby catalyzing pioneering discoveries and innovative breakthroughs in biochemistry, molecular biology, and related disciplines.

## Figures and Tables

**Figure 1 ijms-24-17135-f001:**
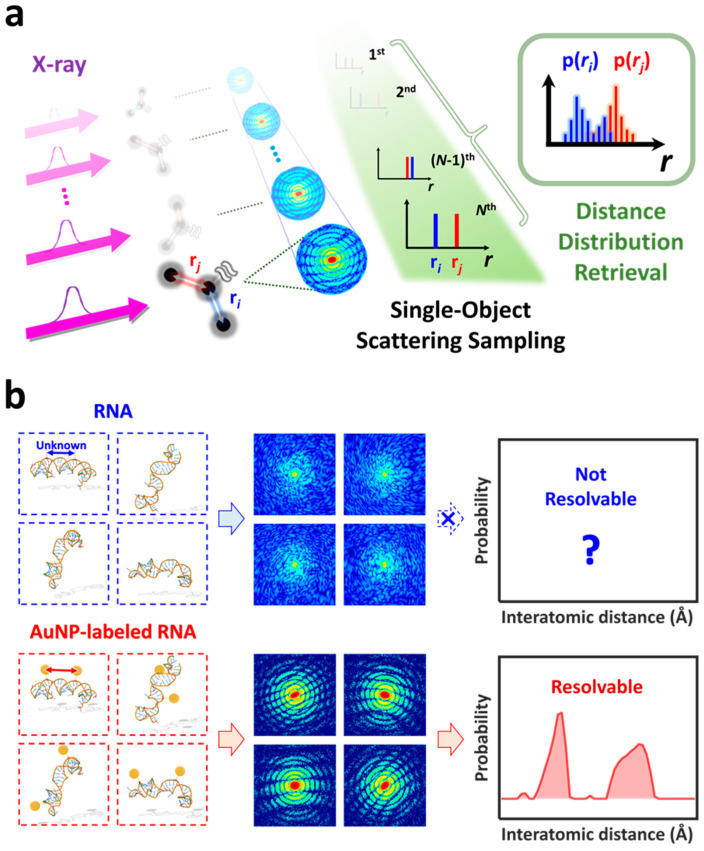
Single-object scattering sampling (SOSS) of biological molecules (Bio-SOSS). (**a**) A conceptual schematic of the single-object scattering sampling (SOSS) technique. The SOSS technique aims to retrieve the instantaneous structure of a single molecule using an ultrashort and intense X-ray pulse. Since the atoms in a molecule scatter the X-ray photons of the incident X-ray pulse, the interatomic distances within a molecule (representatively denoted by r*_i_* and r*_j_*) can be retrieved per each X-ray scattering image (circular pattern). Consequently, *N* instantaneous conformations within the phase space of a molecule correspond to *N* pairs of (r*_i_*, r*_j_*), resulting in a distribution for interatomic pair distances along the r axis. (**b**) An application of SOSS to a biological molecule (Bio-SOSS). (**Top**) A biological molecule, such as RNA, has a flexible structure with different conformations (blue-dotted square). Since the biomolecules mostly consist of light atoms, the X-ray scattering intensity from each conformation (images in the middle column) is too low to accurately resolve the distribution of interatomic distances. (**Bottom**) If the gold nanoparticles (AuNPs) are labeled to two sites (for example, nucleosides in RNA) of interest, the X-ray scattering images from the AuNP-labeled RNA exhibit distinctive patterns depending on the distance and orientation between the two AuNPs (images in the middle column). Thus, we can reconstruct the distribution of structural conformations by parameterizing the distances between the two AuNPs (red curve in the right column).

**Figure 2 ijms-24-17135-f002:**
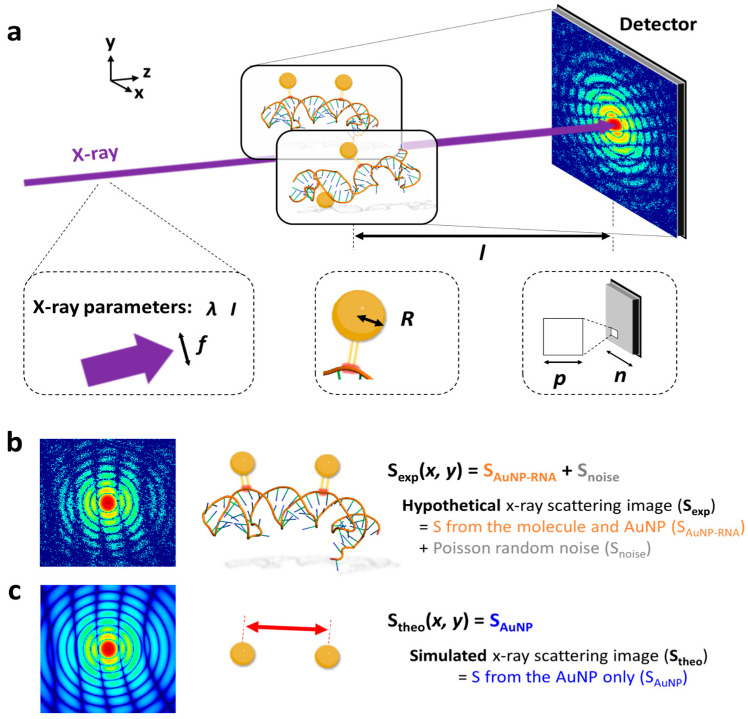
Bio-SOSS simulation. (**a**) A schematic diagram of the Bio-SOSS experiment. In the experiment, the incident ultrashort, ultraintense X-ray probe pulse was spatially focused to a well-defined position where the biomolecule of interest (ssRNA in this work) labeled with gold nanoparticles (AuNPs) was located. The scattered X-ray signal from the molecule was recorded on the 2D area detector placed at *l* away from the sample position. We defined the z-axis as the direction of incident X-ray, and the x- and y-axes as the two unit vectors constituting the detector plane perpendicular to the X-ray propagation. The simulated X-ray scattering images depend on several controllable parameters related to the properties of the X-ray, AuNPs, and detector. The focal size, wavelength, and direct beam intensity of the X-ray are represented as *f*, *λ*, and *I*, respectively (dotted square on the left). The radius of the AuNP used for the labeling is expressed as *R* (dotted square in the center). The size of a single pixel and number of pixels in the detector are denoted as *p* and *n*, respectively (dotted square on the right). (**b**) (**Left**) The mock X-ray scattering image from the experiment (S_exp_). (**Right**) S_exp_ consists of the scattering signals from the single molecule of AuNP-labeled ssRNA (S_AuNP–RNA_), depicted by a molecule in the center, and the random noise (S_noise_). (**c**) (**Left**) The X-ray scattering image of the Bio-SOSS algorithm (S_theo_). (**Right**) S_theo_ took the scattering from the pair of the two AuNPs (S_AuNP_), depicted by a pair of two AuNPs in the center, into consideration. The distance between the two AuNPs were optimized to generate S_theo_, which best reproduces S_exp_. A pair of sample images for S_exp_ and S_theo_, selected from the images for S_RNA_, S_AuNP_, S_AuNP-RNA_, S_noise_, S_exp_, and S_theo_ (Appendix A), is visualized on the left. We note that the scattering intensity was logarithmically scaled to depict the weak signals in the high q (Appendix A). Details for the generation of simulated images are explicitly described in the Appendix A.

**Figure 3 ijms-24-17135-f003:**
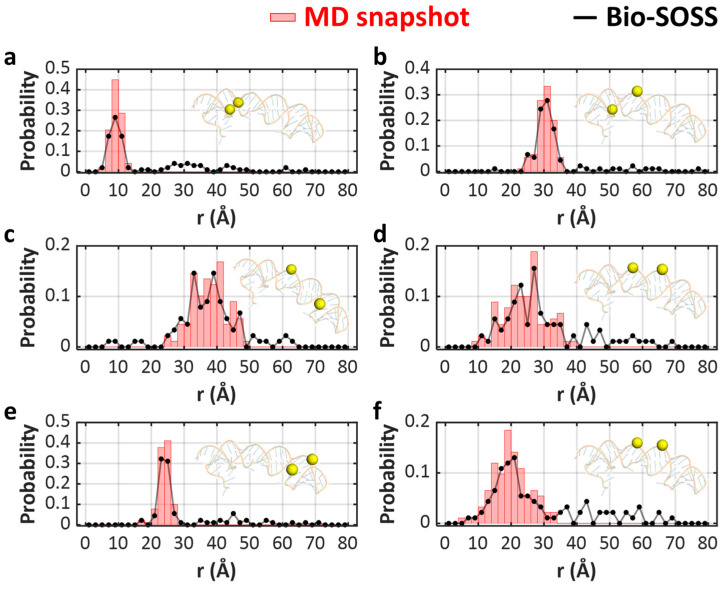
Bio-SOSS reveals the distance distribution between selected pairs of nucleosides. (**a**–**f**) The distribution of retrieved distances between each of the six selected pairs of AuNPs that are substituted to the oxygen atoms of the phosphate group (black dotted line) and the target distribution from MD snapshots (red histogram). In each panel, the ssRNA is schematically shown with the labeling sites indicated with yellow spheres. We implemented the Bio-SOSS to retrieve the distance between AuNP-labeled sites at each MD snapshot. The reconstructed distribution of distances between each pair of AuNPs throughout the 100 MD snapshots were sorted into histograms and compared with the real distribution computed directly from the molecular structures in the MD snapshots. The notable consistency between these two sets of results underscores the potential of Bio-SOSS in capturing the diverse conformations of a RNA molecule. For visualization purposes, we randomly selected six pairs of AuNPs, denoted as (**a**–**f**). We labeled the AuNP at the following residues of the target RNA structure: (**a**) (12nd, 13rd), (**b**) (12nd, 16th), (**c**) (15th, 47th), (**d**) (15th, 50th), (**e**) (28th, 47th), and (**f**) (16th, 50th). The remaining 12 pairs are presented in Appendix A.

**Figure 4 ijms-24-17135-f004:**
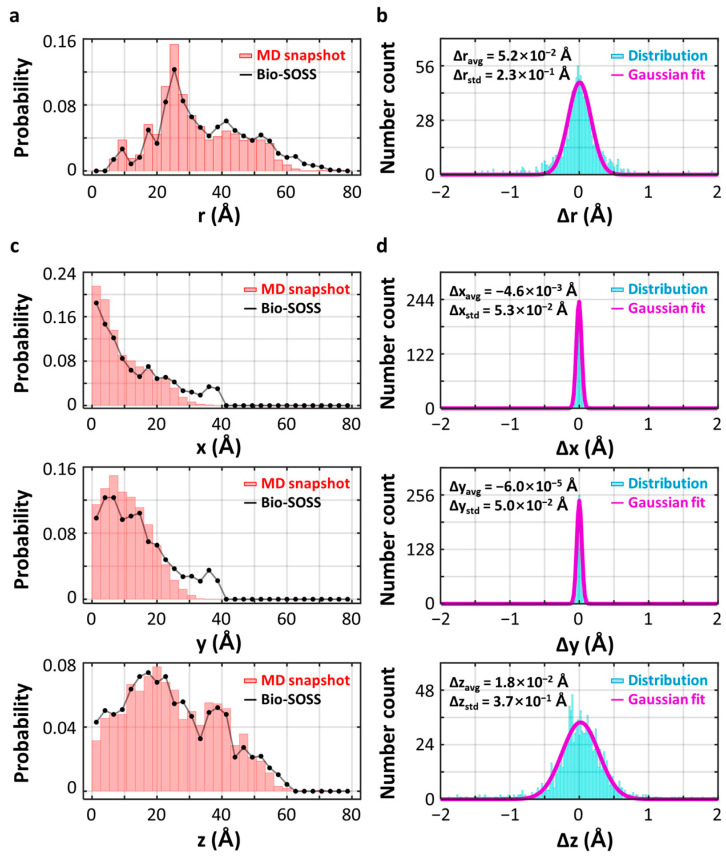
Statistical accuracy of Bio-SOSS. (**a**) A comparison between the distributions of distances between two AuNPs obtained from the analysis of single-object X-ray snapshots (r_fit_, depicted by the black line) and the actual distances between the two AuNPs (r_AuNP_, depicted by the red vertical bars). For a statistical visualization, the distances shown in this diagram were merged from a collection of 18 available pairs of AuNP-labeling sites in the ssRNA strand. The vertical axis represents the probability, or the number frequency of each bin normalized by the total number count. (**b**) The distribution of difference distances between r_fit_ and r_AuNP_, |r_fit_ − r_AuNP_|, was analyzed by fitting it to a Gaussian distribution function. The average (Δr_avg_) and standard deviation (Δr_std_) of the distribution of differences, derived from the fitting results, were 0.052 Å and 0.23 Å, respectively. These extracted values were considerably smaller than typical interatomic bond lengths, demonstrating a remarkable accuracy of Bio-SOSS. The distribution of (**c**) distances and (**d**) difference distances between the AuNPs and AuNP-labeled sites were shown. Both distributions were decomposed into three Cartesian unit vectors: x, y, and z. The average and standard deviation of the difference distance distribution along each axis are presented in the inset of (**d**). The above indicates that the Bio-SOSS method offers higher resolution when measuring distances projected onto the xy-plane (perpendicular to the incident X-ray) compared to distances along the z-axis (parallel to the incident X-ray). We note that the Bio-SOSS simulation in this figure assumes the following experimental conditions: *R* = 9 Å, *n* = 250 pixels, *p* = 234 μm, *I* = 10^14^, *l* = 40 mm, and *f* = 100 nm.

**Figure 5 ijms-24-17135-f005:**
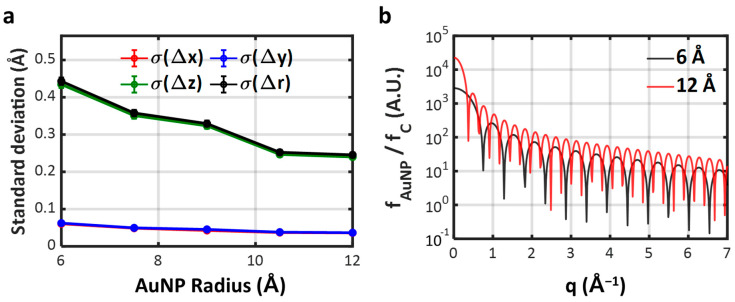
Dependence of the Bio-SOSS accuracy on the size of AuNPs. (**a**) The standard deviation of the distance differences depicted as a function of the radius of AuNP, *R*. The plot shows that the accuracy of the distance retrieval improves with larger AuNPs. (**b**) Dependence of the AuNP form factor on *R*. The AuNP form factors were normalized with the atomic form factor of carbon to show the relative importance of the labeled AuNP compared to those from the RNA chain. The form factors were calculated along the diagonal axis of the detector plane and converted to the momentum space (q). We note that the relative form factors of AuNPs generally decrease in smaller AuNPs, contributing less to the overall X-ray scattering signal, thereby reducing the accuracy of Bio-SOSS.

**Figure 6 ijms-24-17135-f006:**
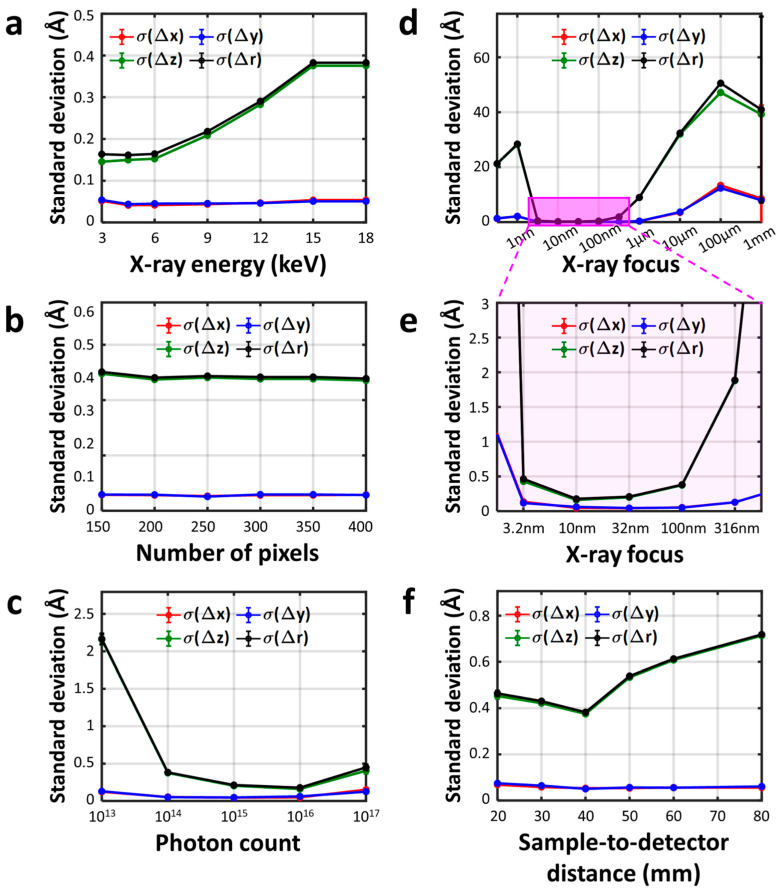
Influence of experimental parameters on Bio-SOSS accuracy. (**a**–**f**) The standard deviation of the distance differences, σ(Δr), visualized as a function of (**a**) X-ray energy, E_x-ray_, (**b**) number of pixels, *n*, (**c**) number of incident X-ray photons, *I*, (**d**,**e**) X-ray focal size at the sample position, *f*, and (**f**) sample-to-detector distances, *l*. In (**a**), an increase in E_x-ray_ leads to a monotonic rise in σ(Δr), while (**b**) shows that σ(Δr) remains constant with varying *n*, highlighting the pivotal role of single-pixel resolution (dq) in Bio-SOSS accuracy, as opposed to maximum q on the detector. The relationship between σ(Δr) and the variables *I* and *f*, as illustrated in (**c**,**d**), implies that there exist optimal conditions for Bio-SOSS with respect to both parameters. Panel (**f**) demonstrates the dependency of σ(Δr) on *l*, unveiling the complex interplay between single-pixel resolution and incident X-ray intensity. Note that (**e**) serves as an expanded view of panel (**d**), emphasizing the trends near the optimal range from *f* = 3.2 nm to 316 nm.

## Data Availability

Additional data supporting the results of this study are available from the corresponding author upon reasonable request.

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
