# Peer review of "Single-Molecule X-ray Scattering Used to Visualize the Conformation Distribution of Biological Molecules via Single-Object Scattering Sampling"

_ijms, 2023, doi:10.3390/ijms242417135_

Round 1
Reviewer 1 Report
Comments and Suggestions for Authors
The manuscript titled “Single-molecule X-ray scattering to visualize the conformation distribution of biological molecules via single-object scattering sampling” by S. Lee, H. Ki, S. J. Lee and H. Ihee describes a method termed as Bio-SOSS that validates through simulations the SOSS methods. This SOSS method is very promising tool to determine the dynamic nature of biomacromolecules. Therefore, the work focused on the computational validation of the results of SOSS of a duplex oligonucleotide. Because of the relevance of this methods, the work deserves to be published but some minor points shall be corrected/completed:
(1) The authors state a ssRNA of 2255 atoms. It will be welcome the description of the number of bases and the motifs of the structure known.
(2) In addition, the reasons for the RNA positions of the AuNP labelling shall be discussed. They just indicate a total of 18 pairs of labelling sites and this is insufficient information.
(3) Have the authors include an open-access tool to run Bio-SOSS in other SOSS experiments?
Reviewer 2 Report
Comments and Suggestions for Authors
This paper presents an innovative technique employing ultrashort X-ray pulses for capturing the instantaneous molecular structure, specifically nuclear acid, through the assistance of interaction with gold nanoparticles (AuNP). This method, termed the Single-Object Scattering Sampling (SOSS) approach, shows promise but could benefit from further testing with additional biological molecules such as peptides, proteins, and various nuclear acids. Please consider the following comments for your review.
1. The title of the paper could either focus specifically on nuclear acid, aligning more closely with the current content, or, if more examples with different biological molecules are included, maintain its broader focus on biological molecules. This would ensure that the title accurately reflects the paper's content.
2. The abstract should concentrate more on the specific work conducted and less on introducing the field. Lines 12-19 of the abstract could be condensed into one or two sentences, while the remaining content could be better suited for the introduction section. Including more detailed descriptions of the work or results in the abstract would enhance its relevance and focus.
3. The paper could explore certain limitations, such as molecular weight or size constraints of this approach. Using nuclear acid as an example, it would be insightful to consider the size of RNA and how this might limit the SOSS approach. The interaction between RNA and AuNP is crucial; thus, exploring whether this can be extended to other RNA sequences with different bases would be valuable.
4. When applying this technique to peptides or proteins, the choice of metal for interaction becomes critical. It's unclear if the existing protocol is suitable for peptides and proteins, and how effectively it would work. The authors could potentially demonstrate this by selecting a typical protein and applying the SOSS approach for a quick test. This would significantly enhance the paper's relevance to a broader range of biological molecules.
5. Line 375 “In light of these considerations, we expect that there must exist an optimal size for labeling particles that strikes a balance between the benefits of larger R (such as the stronger contribution to the X-ray scattering signal as depicted in Figure 5), and smaller R (such as the smaller distortion effects on the RNA chain).” Regarding the confusion around line 375, which discusses an optimal size for labeling particles, a more detailed explanation would be helpful. The balance between larger and smaller radii (R) and their respective effects on the X-ray scattering signal and RNA chain distortion needs clarification.
Round 2
Reviewer 2 Report
Comments and Suggestions for Authors
The authors have addressed all of my concerns. Thanks.